# Pathophysiology of Depression: Stingless Bee Honey Promising as an Antidepressant

**DOI:** 10.3390/molecules27165091

**Published:** 2022-08-10

**Authors:** Fatin Haniza Zakaria, Ismail Samhani, Mohd Zulkifli Mustafa, Nazlahshaniza Shafin

**Affiliations:** 1Department of Neuroscience, School of Medical Sciences, Universiti Sains Malaysia, Health Campus, Kota Bharu 16150, Malaysia; 2Faculty of Medicine, Universiti Sultan Zainal Abidin (UniSZA), Medical Campus, Jalan Sultan Mahmud, Kuala Terengganu 20400, Malaysia; 3Department of Physiology, School of Medical Sciences, Universiti Sains Malaysia, Health Campus, Kota Bharu 16150, Malaysia

**Keywords:** depression, inflammation, monoamine, neurotrophin, stingless bee honey

## Abstract

Depression is a debilitating psychiatric disorder impacting an individual’s quality of life. It is the most prevalent mental illness across all age categories, incurring huge socio-economic impacts. Most depression treatments currently focus on the elevation of neurotransmitters according to the monoamine hypothesis. Conventional treatments include tricyclic antidepressants (TCAs), norepinephrine–dopamine reuptake inhibitors (NDRIs), monoamine oxidase inhibitors (MAOIs), and serotonin reuptake inhibitors (SSRIs). Despite numerous pharmacological strategies utilising conventional drugs, the discovery of alternative medicines from natural products is a must for safer and beneficial brain supplement. About 30% of patients have been reported to show resistance to drug treatments coupled with functional impairment, poor quality of life, and suicidal ideation with a high relapse rate. Hence, there is an urgency for novel discoveries of safer and highly effective depression treatments. Stingless bee honey (SBH) has been proven to contain a high level of antioxidants compared to other types of honey. This is a comprehensive review of the potential use of SBH as a new candidate for antidepressants from the perspective of the monoamine, inflammatory and neurotrophin hypotheses.

## 1. Revisiting Depression

Depression is a psychiatric disorder characterized by psychological, behavioral and physiological symptoms that include a persistent low mood, marked loss of pleasure in most activities, poor concentration, disruptions in appetite and sleeping patterns, cognitive impairments, feelings of worthlessness, excessive guilt, and suicidal thoughts [1]. It is the leading cause of disability worldwide that poses a high emotional and financial burden [2,3]. The World Health Organization (WHO) estimated that depression will be declared a global burden by the year 2030 [4] affecting an estimation of 300 million people from all age categories [5].

Depression covers various subtypes and etiologies [6] from monoamines to inflammatory and neurotrophic propositions. In the 1960s, the “catecholamine hypothesis” appeared as a popular monoamine hypothesis for explaining depression development. It suggested that serotonin (5HT) deficiency and noradrenaline (NA) creates depression [7,8,9]. The inflammatory hypothesis proposes that depression is caused by the interaction of inflammatory cytokine with the hypothalamic–pituitary–adrenal (HPA) axis, consequently affecting the synthesis and reuptake of neurotransmitters [10,11], which subsequently triggers glucocorticoid resistance, glutamate excitotoxicity, and the reduction of brain-derived neurotrophic factor (BDNF) expression [12].

Since BDNF is reduced in the onset of depression, the neurotrophic hypothesis has become one of the critical etiologies of antidepressant progression. This hypothesis states that neurotrophic factors are essential to the development of neurons by promoting synaptic growth and maintaining neuronal survival. They play a crucial role in neuronal network formation and plasticity. On the contrary, the reduction of neurotrophic factors is implicated in the atrophy of stress-vulnerable hippocampal neurons, such as depression and cognitive disorder [13]. This deficiency is believed to be reversed by antidepressant treatments that contribute to the resolution of depressive symptoms [14].

Since the number of depression cases is increasing day by day, the discovery of new treatments is imperative. At present, there is a vibrant demand for new treatment strategies since the flaws of conventional treatments are striking. For instance, many sources purport that antidepressants have a therapeutic delay onset, taking weeks rather than days to become effective [15,16]. Prolonged exposure to antidepressant drugs imposes susceptibility to adverse side effects, such as interferences in sexual functioning, gastrointestinal disturbances, altered sleep pattern, and weight gain [17,18,19,20,21,22]. Moreover, 30% of patients have been reported to be non-compliant with currently available treatments [23,24,25]. Thus, there is a dire need for the development of new antidepressant treatments with better efficacy and that are safer for patients [26].

This has caused an urgent call for complementary and alternative medicines in treating depression [27]. Honey, which contains a variety of active compounds beneficial to brain regulation and treats emotional and psychological disorders including depression [28,29,30] is one of the natural products serving as an alternative medicine [31]. Among the various types of honey, here we focus on stingless bee honey (SBH). In Malaysia, SBH is well known as “madu kelulut” [32]. In addition to SBH, there are other honeys capable of treating several health problems named Tualang and Manuka [33,34]. However, in terms of nutritional composition, SBH contains a higher level of polyphenol [35,36,37], an important active compound that participates in modulating signaling pathways, thus influencing neuronal survival and cell regeneration and development, which suffer detrimental effects after injury [38,39]. To date, there are limited studies highlighting the potential of SBH as an alternative supplement to treat depression. Therefore, this review discusses the different hypotheses associated with depression and how SBH’s mechanism of action could act as a potential antidepressant as a brain supplement. We highlight the different types of etiology hypotheses in the pathophysiology of depression followed by its mechanism of action.

## 2. Pathophysiology of Depression

### 2.1. Monoamine Hypothesis

Depression is a well-known psychiatric disorder that involves the dysregulation of the monoamine system that leads to an imbalance of neurotransmitters, such as 5HT, dopamine, and NA [40,41]. Monoamines are molecules involved in information transmission processes by connecting presynaptic to postsynaptic neurons [42]. They are classified according to their chemical structure and mechanism of action [43]. Since they have different chemical structures, every monoamine is specific to its respective receptors [42] and has a different function in the brain [40,44,45]. For example, 5HT is a central nervous system monoamine that has a crucial role in regulating appetite, circadian cycle, anxiety, memory, and learning. In addition to 5HT, dopamine is another important monoamine that fuels motivation and modulates pleasure, reward, and emotion. In addition, NA is another essential monoamine responsible for attentiveness, emotions, cognition, and social interactions.

The monoamine hypothesis was formulated in the mid-1960s due to the underactivity of brain monoamines such as serotonin, dopamine, and NA in patients’ brains [46]. This hypothesis is based on antidepressant drug efficacy, such as selective serotonin reuptake inhibitors (SSRIs), norepinephrine–dopamine reuptake inhibitors (NDRIs), tricyclic antidepressants (TCAs), and monoamine oxidase inhibitors (MAOIs) [47,48]. The mechanisms of action for this hypothesis with antidepressants are: (1) inhibition of the reuptake of 5HT and/or NA; (2) antagonistic presynaptic inhibition of 5HT and/or NA; and (3) inhibition of monoamine oxidase (MAO) [45]. Findings on these mechanisms of action showed that chronic treatment with antidepressants ultimately causes increased levels of monoamines.

Apart from 5HT, dopamine, and NA, γ-aminobutyric acid (GABA) is also reported to affect depression [49,50,51,52]. GABA plays a role in depression and anxiety through its interaction with inflammatory cytokines, NF-kB, and p38 MAPK signaling pathways [53].

### 2.2. Inflammation Hypothesis in Depression

The inflammation theory has also been linked to depression, which surprises many people. It acts as a key point regarding treatment direction for depression cases. Believed to be fueled by lifestyle, the inflammatory process is related to the nuclear factor-κB (NF-κB) pathway [54], a transcriptional factor that regulates various gene expressions. It is activated by extracellular stimuli, such as lipopolysaccharide (LPS), administration, or chronic stress [55,56,57,58], giving it a propensity to go haywire. Once it is activated, an inflammatory response takes place [59,60].

An inflammatory response includes the secretion of cytokines, which have a specific effect on the interactions and communications between cells [61]. Cytokines are signaling proteins secreted in response to the immune system’s activation by stressors, such as injury, infection, or psychosocial factors [62]. Moreover, the cytokines induce anti- or pro-inflammatory responses, whereby the anti-inflammatory cytokines are secreted to counteract the pro-inflammatory cytokines [63,64]. Cytokines comprise lymphokine (cytokine made by lymphocytes), monokine (cytokine made by monocytes), chemokine (cytokines with chemotactic activities), and interleukin (cytokines made by one leukocyte and acting on other leukocytes). Part of them is recognized as IL-2, IFN-γ, IL-1β, TNF-α, IL-6, IL-12, IL-15 for pro-inflammatory functions [65,66,67] and as IL-4, IL-5, IL-13, IL-1Ra, IL-10 for anti-inflammatory action [68].

Inflammatory responses play a primary role in eliminating or inactivating harmful entities or damaged tissues in the body. However, the over-activation of this system can cause detrimental effects, such as depressive-like behavior [69,70]. Previous studies have shown that depressed people have increased levels of inflammatory mediators, such as C-reactive protein (CRP) and pro-inflammatory cytokines [71,72]. In response to inflammation, the translocation of inflammatory mediators interferes with neuronal and glial well-being, resulting in cognitive and behavioral manifestations, and synaptic plasticity that leads to neurodegeneration [73].

There are two major pathways for inflammatory cytokines that disrupt the synthesis of monoamine neurotransmitters, particularly 5HT, glutamate, and dopamine, as shown in Figure 1. 

They are important for neurotransmitter regulation and ultimately affect mood regulation in depression, namely kynurenine and tetrahydrobiopterin (BH4) [66,74,75]. Activation of the kynurenine pathway (KP) within areas of the brain, such as the hippocampus, has been shown to cause alterations in emotional behaviors [76,77,78]. This is because KP affects the most important neurotransmitter for the regulation of emotion, which is 5HT [79]. When inflammation occurs, levels of indoleamine 2, 3-dioxygenase (IDO) and tryptophan 2, 3-dioxygenase (TDO) are elevated and the tryptophan is used by the IDO and TDO in kynurenine production [80]. This eventually causes the depletion of the tryptophan level for 5HT production. This has been proven in animal models and drug therapy patients with interferon-α [81,82]. IDO and TDO are induced by pro-inflammatory cytokines, such as IL-1, IL-2, IL-6 and IFN-γ [80]. KP causes the increased production of several harmful metabolites, such as 3-hydroxykynurenine (3HK) and quinolinic acid (QA), causing the over-activation of the N-methyl-D-aspartate (NMDA) receptor and inducing oxidative stress and kynurenic acid [83]. The link between inflammation and KP is evident through the increased number of astrocytes that are synthesized by kynurenic acid and the increased production of quinolinic acid by microglia [79]. Alongside kynurenine, the tetrahydrobiopterin (BH4) pathway is also significant due to the monoamine neurotransmitter synthesis that is disrupted in depression [67]. Analyzed SBH sample identified compounds such as phenylalanine, alanine, tyrosine, valine, acetate, lactate, trigonelline, ethanol metabolites, glucose, fructose, sucrose, and maltose [84]. Phenylalanine, which is consistently found in SBH, converted to tyrosine, which simultaneously converts BH4 to 4a-Hydroxytetrahydrobiopterin and is catalyzed by phenylalanine hydroxylase [85]. BH4 is a cofactor for precursors of neurotransmitters, namely 5HT, dopamine, and NA [75]. For example, the serotonergic pathway biosynthesis of 5HT comes from tryptophan, whereas dopaminergic, noradrenergic, and adrenergic pathways are intermediated by the precursor L-3,4-dihydroxyphenylalanine (L-DOPA) for the synthesis of dopamine, adrenaline, and NA [86,87]. Inflammatory cytokines can disrupt BH4 production, which is crucial for neurotransmitter synthesis [67]. There are two mechanisms that are involved in the disruption of BH4. Firstly, inflammatory cytokines stimulate NOS to produce NO. The elevated activity of NOS causes the increased utilization of BH4 that will be converted to 7, 8-dihydrobiopterin (BH2).

The conversion of arginine to nitric oxide (NO) by nitric oxide synthase (NOS) is enhanced by BH4, which acts as an enzyme co-factor [88]. Furthermore, BH4 is very sensitive to oxidative stress. Inflammatory cytokines are known to increase oxidative stress through the production of both nitrogen and oxygen-free radicals. This causes the irreversible degradation of BH4 to dihydroxyanthopterin [89].

### 2.3. Neurotrophin Hypothesis

In addition to the monoamine and inflammatory hypothesis, the neurotrophin hypothesis also has a vital role in the pathophysiology of depression [90]. Neurotrophin is a type of protein that is essential for the growth, survival, and differentiation of neurons [91,92]. Four types of neurotrophins are important in mammals, namely brain-derived neurotrophin factor (BDNF), nerve growth factor (NGF), neurotrophin-3 (NT-3), and neurotrophin-4 (NT-4) [93]. BDNF is critical in the central nervous system (CNS) for neurogenesis, synaptic plasticity, development, survival, and neuron maintenance [94,95,96,97]. BDNF is an example of a neurotrophin that has an impact on the pathophysiology of depression [13,97,98]. BDNF and its receptor tropomyosin receptor kinase B (TrKB) are involved in different intracellular signaling pathways, such as mitogen-activated protein kinase/extracellular signal-regulated protein kinase (MAPK/ERK), phospholipase Cγ (PLCγ), and phosphoinositide 3-kinase (PI3K) [99]. These pathways have a biological impact on the central nervous system, such as on memory and mood regulation [95,100,101]. ERK is one of the downstream BDNF pathways that is implicated in the regulation of mood and behavior in the depression model that mediates the effects of antidepressants [102,103,104,105]. Meanwhile, PI3K signaling is an important component of long-term potentiation (LTP) [106]. This signaling pathway acts as a biochemical cascade for α-amino-3-hydroxy-5-methyl-4-isoxazolepropionic acid receptor (AMPAR) for synaptic plasticity, thus causing behavioral alteration [107]. Moreover, all the intracellular signaling pathways that were mentioned earlier have been discussed in previous studies that are related to depression. Changes in BDNF levels in the central nervous system disrupt the entire signaling pathway, which can lead to various psychological disorders, including depression [108,109,110,111,112].

BDNF promotes neurogenesis, which is part of neuroplasticity. Neuroplasticity involves changes or alterations in the structure, functions, and connections of the central nervous system (CNS) in response to intrinsic or extrinsic stimuli [113]. These changes include the morphology of mature neurons, such as axonal and dendritic arborization and pruning, increased spine density, and synaptogenesis [114]. Neurogenesis is defined as the formation of newborn neurons in proliferative areas that include the subventricular zone (SVZ) and the subgranular zone (SGZ) of the dentate gyrus region in the hippocampus area [114,115]. This region is crucial for memory, learning, and other cognitive functions [116]. The alteration of BDNF levels is known to be detrimental to neurogenesis in the hippocampus, specifically in the dentate gyrus region [117]. The dentate gyrus is a region in the hippocampus that is widely discussed in depression [118,119,120]. Based on previous studies, BDNF levels in the hippocampus and prefrontal cortex are reduced in cases of depression [13,121,122]. This consequently resulted in the decreased size of the hippocampal area in the brain in both clinical and preclinical studies of depression [118,119,120]. Similarly, the condition is observed in the prefrontal cortex [118,123,124] causing neuronal loss and synaptic dysfunction in cortical limbic regions that ultimately disrupt mood and emotions [73].

In addition to neurogenesis as a part of neuroplasticity related to BDNF in the brain, synaptic plasticity is also associated with depression [41,125,126]. Synaptic plasticity is essential for the physiological morphology of neurons, and BDNF is one of the crucial regulators in this process making it a therapeutic target in depression [127]. Therefore, the BDNF level is vulnerable to synaptic plasticity in the brain. Long-term potentiation (LTP) is the main mechanism that mediates neuroplasticity at a functional level; synaptic strength is crucial for the connection between neurons in the brain [113]. BDNF facilitated LTP in the Schaffer collaterals of the hippocampus in a young animal model by inducing the release of presynaptic neurotransmitters [128]. Furthermore, the postsynaptic release of BDNF induces LTP in the dentate gyrus [129]. Increased hippocampal dendritic spine by LTP is contributed by BDNF signaling together with local protein translation [130]. Based on previous studies, patients showed decreased hippocampal volume and BDNF expression during depressive episodes compared to patients in remission, which altered synaptic plasticity by elevating hippocampal dendritic atrophy and cell death contributing to the decline of LTP [122,131,132]. These features have also been observed in rodents [133,134,135]. Moreover, the reduction of LTP caused by depression has been observed especially in the hippocampus and the prefrontal brain area [73,136,137]. The same effect has been observed in long-term depression (LTD) [138].

## 3. Stingless Bee Honey (SBH) as an Antidepressant

Earlier in this review, the etiology of pathological depression was discussed briefly. In this section, the role of SBH as a prophylactic against depression is reviewed. The focus will be on certain properties of SBH that are related to depression, which include amino acid (phenylalanine), antioxidant properties, and anti-inflammatory effects. 

### 3.1. Neurotrophic Factors

The complex biological properties of SBH consist of amino acids (phenylalanine, alanine, tyrosine, and valine), phenolic compounds, carbohydrates, organic acids, vitamins, minerals, lipids, and enzymes [139,140,141,142,143]. They have potential roles in the regulation of signaling pathways in depression development. The amino acid that is highlighted in this review is phenylalanine. Phenylalanine is an essential amino acid that needs to be ingested through diet since it is not naturally synthesized by the body [144]. It is an important amino acid for the synthesis of neurotransmitters and a precursor for dopamine and NA [87]. This can be related to the role of SBH as a prophylactic against depression, which is in line with the monoamine hypothesis. The monoamine hypothesis, which was explained earlier, stated low levels of neurotransmitters in depressed patients as well as in animal studies. Furthermore, the role of NA has been emphasized in attenuating microglial activation in the brain, thus inhibiting pro-inflammatory cytokines (inflammatory hypothesis) as well as enhancing the production of neurotrophic factors (neurotrophin hypothesis) for neurogenesis in the brain [145]. Moreover, neurotransmitters have been reported to enhance BDNF release in the brain [146]. Therefore, neurotransmitters and BDNF have a bilateral effect that can ameliorate depressive behavior. This indicates that the administration of SBH during depressive episodes could regulate the deficiency of neurotransmitters as well as BDNF, which is important to the neurological process. The regulation of the neurological process during depressive episodes would impede depressive behavior symptoms, such as sickness behavior, loss of motivation, and anhedonia [147,148,149].

### 3.2. Antioxidant

Stingless bee honey (SBH) has high levels of phenolic compounds compared to other honey [36,37,150]. Examples of phenolic acids in SBH are p-coumaric acid, gallic acid, caffeic acid, chrysin, and apigenin [28,150,151,152]. Antioxidant properties in honey have been reported to strongly correlate with phenolic compounds [153,154,155]. The color intensity of honey is also an indicator of antioxidant activity due to the presence of pigments such as carotenoids and flavonoids [156]. According to Kek and colleagues (2014), SBH has higher color intensity compared to Tualang, Gelam, Pineapple, Borneo, and commercial honey [36]. Antioxidants are important as scavengers in preventing oxidative stress that leads to DNA damage [157,158]. The brain is a highly susceptible organ to the elevation of oxidative stress due to its high oxygen demand [48,159,160]. Several researchers have reported the neuroprotective effect that resulted from the polyphenol content in honey [29]. These studies support SBH as a potential antidepressant since depression is also related to oxidative stress within the brain [41,161,162]. According to a study by Czarny and colleagues (2018), depressed patients had elevated reactive oxygen species (ROS) and nitrogen species (RNS) from oxidative DNA damage after depressive episodes [163].

Moreover, the relationship between oxidative stress and depression has been reviewed and discussed relating to its usage as a natural compound with antioxidant properties as a constituent of their polyphenols that can alleviate depression [41,164]. For example, antioxidant activity by p-coumaric acid has been identified in animal models of depression [48]. It has also been reported to show a neuroprotective response through increased levels of glutathione and superoxide dismutase that subsequently reduce oxidative stress capacity and neurotoxicity [165,166,167]. In addition to p-coumaric acid, chrysin also exhibited oxidative stress in the preclinical model of depression [168,169]. The properties of phenolic compounds proven to show efficacy based on both animal and human studies are displayed in Table 1.

The antioxidant activity in SBH can act as ROS scavengers that can regulate the mechanism of action to improve depression in patients. Antioxidants can cause prophylactic effects in depression since elevated ROS have been reported to affect BH4. In the inflammatory hypothesis, BH4 is briefly discussed as a cofactor for the biosynthesis of neurotransmitters 5HT, dopamine, and NA. The regulation of BH4 after the administration of SBH regulates neurotransmitters that consequently enhance mood, emotion, and behaviors after depressive episodes. Neurotransmitters are known for their role as modulators in brain activity. In addition, SBH possesses high intensity in terms of color pigment indicating the presence of flavonoids. Flavonoids found in honey activate the ERK and protein kinase B (PKB/Akt), leading to the activation of cAMP response element-binding protein (CREB), a transcription factor responsible for increasing BDNF expression [161]. Specifically, a gene expression analysis showed that BDNF and Itpr1 were affected following SBH treatments, and the results indicate that supplementation with SBH leads to specific upregulations of the gene expression tested and leads to an improvement in the depressive state [84]. There are also reports about the use of flavonoids in neurodegenerative disorders to regulate BDNF [189,190,191,192]. This evidence strongly suggests that SBH, which contains a high number of flavonoids, has the ability to enhance BDNF and regulate its levels as a determinant for antidepressant efficacy [91,93,193,194].

### 3.3. Anti-Inflammatory

Stingless bee honey (SBH) has been reported to possess anti-inflammatory properties [150,195]. According to a study by Ranneh and colleagues (2019), SBH has an anti-inflammatory effect in the LPS animal model. The etiology of depression has been discussed in depth through the inflammatory hypothesis perspective, especially with regard to how inflammation affects depression. Inflammation is known to activate the kynurenine pathway in depression [196]. This pathway causes cascade effects through the elevation of ROS and glutamate toxicity as well as a reduction of 5HT and BDNF [197,198,199]. Due to its anti-inflammatory properties, SBH can also regulate BDNF, hence alleviating depression symptoms. This is also essential due to its bilateral effect on the regulation of monoamine systems, which is important in ameliorating symptoms of depression [14,200].

The immune system is very sensitive to oxidative stress and with moderate exercise, immune functionality can be enhanced [48]. Exercise also has been recognized as a useful non-pharmacological strategy to improve the treatment of depression [201,202]. Relaxation responses significantly reduced the neuropsychological scores tested, decreased cortisol, decreased the trend of NGF, and increased BDNF levels [201]. BDNF binds to the tyrosine kinase β receptor (TrKβ) and activates the phosphoinositide 3-kinase and Akt pathway, which inhibits the activity of glycogen synthase kinase-3 beta (GSK-3β) [202,203]. GSK-3β activity cleaves cadherin–β-catenin binding; therefore, GSK-3β inhibition stabilizes β-catenin, which regulates gene expression, synaptic plasticity, and neurogenesis, which in turn has antidepressant effects. A study using flavonoids found they seem to exert additional positive effects with exercise, where a combination of quercetin and exercise training exerted potent anti-tumor and anti-depressive effects through the suppression of inflammation and the upregulation of the BDNF/TrKβ/β-catenin axis in the prefrontal cortex of 1,2-dimethylhydrazine (DMH)-induced colorectal cancer-induced rats [202]. Thus, it is speculated that flavonoids in SBH express anti-inflammatory properties and are involved in the BDNF/ TrKβ pathway; however, the specification of active possible chemotherapeutic modality needs further investigation. This statement supports the potential of SBH as an antidepressant due to its anti-inflammatory properties.

Recent studies on treatment resistance in depressed patients have related the condition to inflammation. Patients with treatment-resistant depression (TRD) have been reported to show dysregulated inflammatory activity compared to non-TRD patients [204,205]. They also exhibited increased levels of inflammatory cytokines (IL-6, IL8, TNF-α, CRP, and macrophage inflammatory protein-1 (MIP)-1 alpha) that resulted in poorer treatment outcomes [206]. Inflammatory cytokines are known as critical mediators in the inflammatory response that disrupt the signaling pathways or mechanisms of action of conventional antidepressants [204]. Therefore, anti-inflammatory treatments might be effective in preventing TRD in patients [207]. 

## 4. Conclusions

In conclusion, stingless bee honey could regulate the detrimental effects during or after depressive episodes that can lead to prophylactic effects. This review summarized how the amino acid (phenylalanine), antioxidant, and anti-inflammatory properties of SBH have the potential to control depression symptoms according to the etiology of depression—the monoamine, neurotrophin, and inflammatory hypotheses—through neurotrophic factors and its antioxidant and anti-inflammatory properties as shown in Figure 2. 

Our review reports the possible mechanisms of stingless bee honey pertaining to its antioxidant and anti-inflammatory properties, which contribute to its antidepressant properties. Follow-up studies are still required to comprehensively analyze the bioactive compound of SBH responsible for the underlying mechanisms as well as to investigate other possible pathways contributing to its anti-depressive effects.

## Figures and Tables

**Figure 1 molecules-27-05091-f001:**
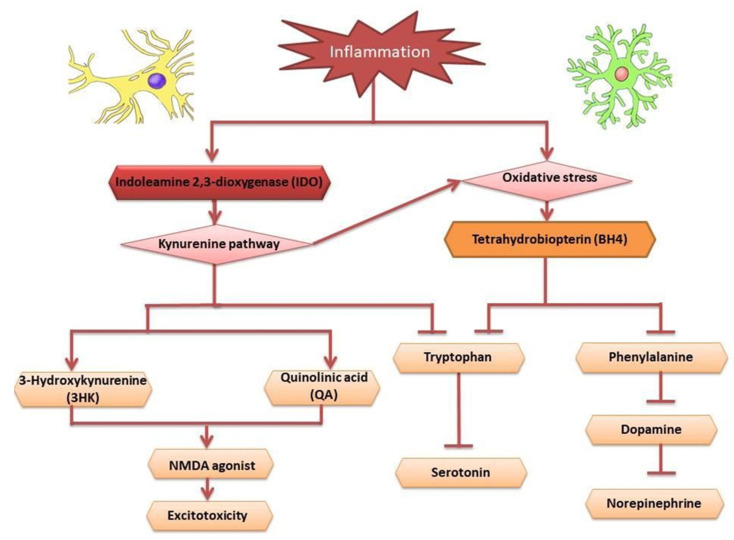
Summary of the mechanism of action for inflammation leading to depression.

**Figure 2 molecules-27-05091-f002:**
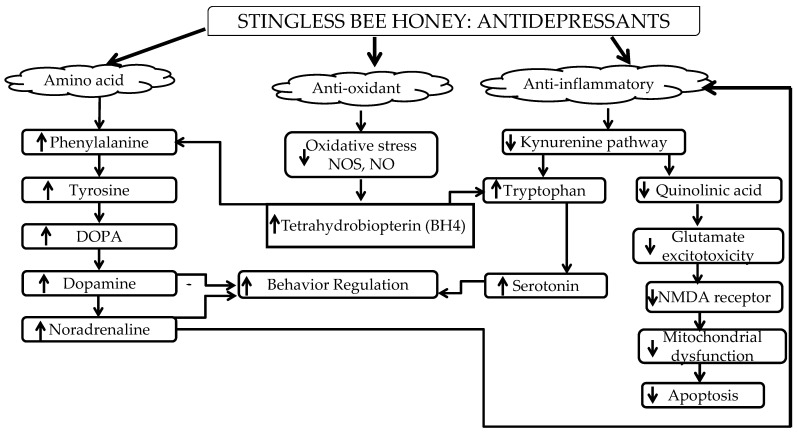
Summary of the anti-depressive effects of SBH.

**Table 1 molecules-27-05091-t001:** Phenolic compounds that showed therapeutic effects on psychiatric and neurological disorders.

Type	Compounds	Therapeutic Effects		
Phenolic acids	p-Coumaric acid 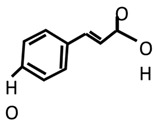	Anti-inflammatory	Antidepressant	Antioxidant
		[49,170,171]	[49]	[171,172,173]
	Gallic acid 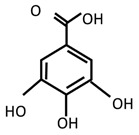	[171]	[174,175,176]	[171,174]
	Caffeic acid 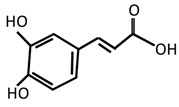	[149]		
Flavonoids	Chrysin 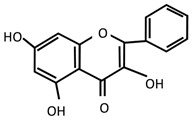	[168,169]	[177]	[168,169,177]
	Apigenin 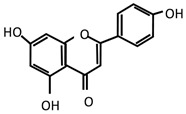	[178,179,180,181,182,183]	[180,181,184,185,186,187]	[180,183,188]

## Data Availability

Not applicable.

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
