# Peer review of "Pathophysiology of Depression: Stingless Bee Honey Promising as an Antidepressant"

_molecules, 2022, doi:10.3390/molecules27165091_

Round 1

Reviewer 1 Report

General Concerns:

This is a descriptive review about the potential use of stingless bee honey (SBH) as a new candidate for antidepressants from the perspective of monoamine, inflammatory and neurotrophin hypotheses. The text is nicely written, and shows the possible mechanisms of SBH to its antioxidant and anti-inflammatory which contributing to antidepressant properties. Some parts of this review need to be refined, as outlined here:

Major:

 i) Recently, molecular and cellular approaches have demonstrated that ROS and antioxidants can directly affect the control of gene expression. In this regards the authors do not state the role BDNF gene expression and the antioxidants.

ii) The authors should explain the role of non-pharmacological factors and neurotrophins expression. For example a recent work, shows that the expression of BDNF and NGF, is also modulated by relaxation methods (see Zappella et al 2021), these new studies have a pivotale role in the modulation of BDNF expression, in the present review these new data are missing. 

Minor:

Line 23  change “this article” with “this review”

 Line 32.  What’s the difference between persistent depression and major depression?

Line 69, point is missing

Line 76, typing error

Line  98, after “(MAOIs) [47, 48]”  put the point

Line 126, typing error

Line 158typing error

Line 209,  typing error

 Standardize the following citations  in the bibliography, please.

N. 7, page number missing

N. 8, point missing

N. 12, why use DOI?

N. 14, typing error

N. 15, pag are missing

N. 18, typing error

N. 21, typing error

Review the follow citations, please:

N. 29

N. 33

N. 37

N. 49

N. 70

N. 71

N. 73

N. 79

N. 121

N 124

N 145

N 152

N 170

N 180

Author Response

Dear Reviewer,

Thank you

Reviewer 2 Report

The manuscript "Pathophysiology of depression: Stingless bee honey promising as an antidepressant" is a novel concept. The paper has scientific merit for publication. There are some minor spell mistakes or typo errors. Authors are requested to correct such inconsistencies.

Author Response

Dear Reviewer,

Thank you

Reviewer 3 Report

1. Although stingless bee honey contains many active components which may be related to the treatment of depression, the levels of components are very important for their anti-depression effects. Thus, the review should describe the levels of related component in stingless bee honey. If the level is too low to cure depression, it does not make sense. Maybe, this is why line 74-75 “To date, there is limited studies highlighted the potential of SBH as 74 an alternative medicine to treat depression”.

2. There are too many grammatical errors in the full manuscript, including but not limited to the below examples,

Line 34, The World Health Organization …

Line 40, serotonin (5HT) deficiency …

Line 41, The inflammatory hypothesis proposes …

Line 47, This hypothesis states …

Check the manuscript and correct the errors.

3. Line 83, 5HT and NA are mentioned and abbreviated in line 40, therefore, correct them in line 83. Also, dopamine is abbreviated as DA, however, DA is not used in the full manuscript, remove DA.

4. Line 95, 5HT replaces serotonin.

5. Line 100, what is NE? give full name.

Author Response

Dear Reviewer,

Thank you

Reviewer 4 Report

This manuscript need alot of refinement. Understanding of the  manuscript is little tough and moreover the machanism needs to ellobrate. Missing links as proper diagram can be provided properly. 

Author Response

Dear Reviewer,

Thank you.

Round 2

Reviewer 1 Report

Standardize the citations in the bibliography, please. 

Reviewer 3 Report

The manuscript was modified well.